# Comparison of the Germination Conditions of Two Large-Spore Microsporidia Using Potassium and Sodium Ion Solutions

**DOI:** 10.3390/insects14020185

**Published:** 2023-02-13

**Authors:** Yuji Imura, Haruka Nakamura, Reina Arai, Yoshinori Hatakeyama

**Affiliations:** Laboratory of Applied Entomology, Department of Agricultural Bioscience, College of Bioresource Sciences, Nihon University, 1866 Kameino, Fujisawa 252-0880, Kanagawa, Japan

**Keywords:** microsporidia, germination condition, *Trachipleistophora*, *Vavraia*, pébrine

## Abstract

**Simple Summary:**

In this study, we investigated the germination conditions of the large-spore microsporidia *Trachipleistophora* sp. FOA-10 and *Vavraia* sp. YGSL-2015-13. The germination rates of the microsporidia were determined using a combination of four different treatment solutions and two test temperatures and were compared to those of *Trachipleistophora* sp. OSL-2012-10 and *Nosema bombycis*, which have known germination conditions. We found that the two types of microsporidia germinated in physiological salt solution. These differences in germination rates were affected by the preservation solution and temperature.

**Abstract:**

The germination of a microsporidian polar tube generally occurs under alkaline conditions. Typically, microsporidian spores can be stored in physiological salt solution for short periods. However, because of differences in the lodging area, the requirements may not always be uniform. In fact, *Trachipleistophora* sp. OSL-2012-10 (*nomen nudum Trachipleistophora haruka*) germinated when preserved in physiological salt solution. In this study, the germination characteristics of the large-spore microsporidia *Trachipleistophora* sp. FOA-2014-10 and *Vavraia* sp. YGSL-2015-13 were compared with those of *Trachipleistophora* sp. OSL-2012-10; moreover, we investigated whether these characteristics are specific to these microsporidia. We found that both microsporidia germinated in physiological salt solution. These differences in germination rates were affected by the preservation solution and temperature.

## 1. Introduction

Pébrine is one of the most important illnesses associated with the breeding of silkworms (*Bombyx mori* L.) [1]. This infectious disease is caused by the infection of silkworms by the microsporidium *Nosema bombycis* Nägeli. The infection of silkworms by microsporidia is called microsporidiosis. Microsporidia are obligate intracellular parasitic fungi. Approximately 40% of microsporidian species infect insects [2,3], with peroral infection being the most common route of infection [1,4,5]. After infecting insects, microsporidia enter the midgut, where they germinate in response to stimulation by alkaline intestinal juice and/or potassium ions [6]. Microsporidian spores germinate in four stages: (i) activation of spores, (ii) increase in intrasporal osmotic pressure, (iii) release and eversion of the polar filament, and (iv) passage of the sporoplasm via the polar filament. However, the germination process of various microsporidian spores remains unclear [7,8]. The process by which spore germination is activated has been elucidated, and its triggers include high or low pH and a pH shift from acidic to alkaline or from alkaline to neutral [9,10]. Moreover, the germination of the spores of several strains of microsporidia requires both alkaline and acidic conditions [11]. Therefore, the germination conditions vary according to the host and external environments [10]. Accordingly, a clarification of the germination conditions of microsporidian spores will lead to an improvement in and more accurate selection of experimental and preservation methods.

In our previous study, we investigated the germination properties of *Trachipleistophora* sp. OSL-2012-10 (*nomen nudum Trachipleistophora haruka*) [6], which represented the first record of the large-spore-sized microsporidia genus of *Trachipleistophora* among insect-infection-causing organisms in Japan [12]. This strain is infectious to silkworms (our non-published data). The silk gland is the primary site of infection, with infected silkworms being unable to vomit the thread. Most entomopathogenic microsporidia germinate in the presence of potassium ions in insects and remain stable in the presence of sodium ions. However, *Trachipleistophora* sp. OSL-2012-10 exhibits specific germination conditions under specific salt conditions, including the presence of sodium or potassium ions, and temperature conditions [6]. Thus, stock solutions of *Trachipleistophora* sp. OSL-2012-10 must be prepared using distilled water (DW). An investigation of the germination conditions of microsporidia in infected silkworms is also essential for their safe breeding. Therefore, we decided to search for microsporidia with similar characteristics that infect other insects.

We reported the detection of microsporidia of nearly the same size as *Trachipleistophora* sp. OSL-2012-10 in a dragonfly (*Orthetrum albistylum speciosum* Uhler; Odonata: Libellulidae) [13] and the common cutworm (*Spodoptera litura* L.; Lepidoptera: Noctuidae) [14]. These large-spore microsporidia have been detected in field insects but not in silkworm rearing rooms in Japan. Thus, it is possible that large-spore microsporidia are introduced into silkworm rearing rooms through feed such as mulberry. Microsporidia with different characteristics can enter silkworm rearing rooms and cause microsporidiosis, which cannot be prevented using conventional methods. Thus, in this study, we investigated the germination properties of two microsporidian strains—*Trachipleistophora* sp. FOA-2014-10 (hereafter, FOA) and *Vavraia* sp. YGSL-2015-13 (hereafter, YGSL)—using a similar method as that reported by Nakamura et al. [6]; moreover, we compared these strains with *Trachipleistophora* sp. OSL-2012-10 (hereafter, OSL).

## 2. Materials and Methods

### 2.1. Microsporidia

We analyzed the germination of the spores of FOA and YGSL (Table 1). FOA was isolated from the adult dragonfly *O. albistylum speciosum* in Kanagawa, Japan in 2014 (Table 1). Its spores had a length of 4.42 ± 0.28 µm and width of 2.35 ± 0.14 µm. This strain was classified into the genus *Trachipleistophora* based on its small-subunit (SSU) rRNA gene sequence [13]. Meanwhile, YGSL was isolated from the common cutworm *S. litura* in Yamaguchi, Japan in 2015. Its spores had a length of 5.36 ± 0.40 µm and width of 2.91 ± 0.20 µm (Table 1). This strain was classified into the genus *Vavraia* based on its SSU rRNA gene sequence [14]. These two microsporidian strains have large spores similar to those of OSL (spore size: length, 4.54 µm; width, 2.66 µm) [12]. We used OSL and *N. bombycis* NIS-001 (hereafter, NIS-001) as the experimental controls because the germination conditions of their spores have been analyzed in detail by several authors [6,15,16,17,18].

The spores of all four microsporidia were stored in suspension in DW at 4 °C. Additionally, the data on the germination rates of OSL and NIS-001 used in this study were quoted from Nakamura et al. [6].

### 2.2. Measurement of Germination Rates

The experimental conditions were set up, and germination rates were measured, as described by Nakamura et al. [6]. We recorded the germination rates in the following treatment solutions: (i) DW, (ii) 0.85% (*w*/*v*) physiological salt solution (FUJIFILM Wako Pure Chemical Corp., Osaka, Japan), (iii) 0.1 M potassium hydroxide (KOH solution, FUJIFILM Wako Pure Chemical Corp.), and (iv) 0.1 M potassium chloride + 3.0% (*w*/*v*) hydrogen dioxide (KCl + H_2_O_2_ solution, FUJIFILM Wako Pure Chemical Corp.). The germination rates were measured at 4 °C and 25 °C.

Meanwhile, 100 μL of the spore suspension solutions in 1.5-mL tubes was incubated at 4 °C or 25 °C for 1 h. For subsequent experiments at 4 °C, the tests were performed in a 4 °C low-temperature room. The spore suspension solutions were centrifuged at 3000 rpm (Centrifuge Eppendorf 5415D, Eppendorf Ag., Hamburg, Germany) for 10 min, and the supernatant was removed. To these 1.5-mL tubes, 1 mL of DW, 0.85% (*w*/*v*) physiological salt solution, KOH solution, or KCl + H_2_O_2_ solution was added, followed by incubation at room temperature for 40 min. The tubes were centrifuged at 3000 rpm for 10 min, and the supernatant was removed. Subsequently, 500 μL of 0.25% (*w*/*v*) trypsin solution (FUJIFILM Wako Pure Chemical Corp.) was added to each tube, followed by incubation at room temperature for 40 min to dissolve the polar tube. The tubes were centrifuged at 3000 rpm for 10 min, and the supernatant was removed. Next, 100 μL of DW was added to resuspend the pellet, following which 6 μL of the spore suspension solutions was applied to a disposable cell counting plate (Neubauer Improved Type, WATSON Co., Ltd., Tokyo, Japan), and the numbers of germinated spores (germinated spores appear black owing to the ejection of cytoplasm) and ungerminated spores were measured using a phase-contrast microscope (×400, OPTIPHOT-2, Nikon Co., Tokyo, Japan). The number of ungerminated spores was divided by the total number of spores and expressed as a percentage. This procedure was repeated three times for each of the four conditions.

### 2.3. Statistical Analysis

Statistical analysis of the relationship between the microsporidian strains and the germination rate was performed using a two-way analysis of variance (ANOVA; with significance set at *p* < 0.05). When statistically significant differences were detected, a Tukey’s test or Bonferroni post hoc test was performed for multiple comparisons (*p* < 0.05). Statistical analysis was performed assuming that normality and homoscedasticity were followed.

## 3. Results

### 3.1. Germination Rates in DW

The germination rates of the microsporidian strains at two different temperatures (4 °C and 25 °C) were investigated using DW (Table 2). The mean germination rates of FOA were 2.0% and 4.1% at 4 °C and 25 °C, respectively. Meanwhile, the mean germination rates at the two temperatures were 2.6% and 22.2%, respectively, for YGSL, 11.4% and 11.8%, respectively, for OSL, and 2.5% and 2.1%, respectively, for NIS-001. Thus, YGSL exhibited the highest germination rate (22.2% at 25 °C) in DW.

Statistical analysis revealed significant differences between strains (*F*_3,16_ = 3.24, *p* = 0.01), test temperatures (*F*_1,16_ = 4.49, *p* = 0.04), and interactions (*F*_3,16_ = 3.24, *p* = 0.03; two-way ANOVA, *p* < 0.05, *n* = 3). Multiple comparisons revealed significant differences in the germination rates of FOA, YGSL, and NIS-001 compared to that of OSL at the two test temperatures (*q* = 16.67, 15.72, and 15.78, respectively), as determined using a Bonferroni test (*q*_4,3_ = 6.82, *p* < 0.05, *n* = 3; Figure 1A). Additionally, the mean germination rate of YGSL differed significantly from those of FOA and NIS-001 at 25 °C (Bonferroni test, *q*_4,3_ = 6.82, *p* < 0.05, *n* = 3; Figure 1A). Significant differences were observed between the test temperatures for YGSL (*q* = 8.09; Tukey’s test, *q*_2,3_ = 4.50, *p* < 0.05, *n* = 3; Figure 1A).

### 3.2. Germination Rates in Physiological Salt Solution

The germination rates of the microsporidian strains were also measured at different temperatures using a physiological salt solution (Table 2). The mean germination rates of FOA were 2.9% and 34.6% at 4 °C and 25 °C, respectively, versus 6.5% and 48.2%, respectively, for YGSL, 14.3% and 31.0%, respectively, for OSL, and 5.4% and 1.6%, respectively, for NIS-001. At 25 °C, all strains but NIS-001 had a mean germination rate exceeding 30%. In particular, YGSL exhibited stable germination rates (55.6%, 50.2%, and 38.8%) in the three trials performed at 25 °C.

Statistical analysis revealed significant differences between the microsporidian strains (*F*_3,16_ = 3.24, *p* < 0.001) and test temperatures (*F*_1,16_ = 4.49, *p* < 0.001; two-way ANOVA, *p* < 0.05, *n* = 3). In multiple comparisons of the germination rates in physiological salt solution, Tukey’s test revealed no significant differences at 4 °C, and only YGSL and NIS-001 displayed significant differences at 25 °C (*q* = 8.76; *q*_4,3_ = 6.82, *p* < 0.05, *n* = 3; Figure 1B). Conversely, significant differences in the germination rate between the test temperatures were found for FOA (*q* = 4.75), YGSL (*q* = 9.93), and OSL (*q* = 5.97; Tukey’s test, *q*_2,3_ = 4.50, *p* < 0.05, *n* = 3; Figure 1B).

### 3.3. Germination Rates in KOH Solution

A KOH solution was used to examine the germination rates of microsporidian strains at 4 °C and 25 °C (Table 2). The mean germination rates of FOA were 1.9% and 10.2% at 4 °C and 25 °C, respectively, compared to 61.6% and 56.5%, respectively, for YGSL and 33.8% and 40.7%, respectively, for OSL. Meanwhile, the mean germination rates of NIS-001, which is known to germinate under alkaline conditions, were 79.6% and 83.2% at 4 °C and 25 °C, respectively. The germination rate of NIS-001 was as high as approximately 80%, regardless of the test temperature, whereas that of FOA was low, ranging from 5.4–18.9% in the three trials performed at 25 °C.

Statistical analysis only revealed significant differences among microsporidian strains (*F*_3,16_ = 3.24, *p* < 0.001; two-way ANOVA, *p* < 0.05, *n* = 3). Multiple comparisons revealed significant differences between FOA and YGSL at 4 °C (*q* = 11.96) and between FOA and NIS-001 at 4 °C (*q* = 15.55; Tukey’s test, *q*_4,3_ = 6.82, *p* < 0.05, *n* = 3; Figure 1C). At 25 °C, the germination rates of YGSL, OSL, and NIS-001 differed significantly from that of FOA (*q* = 14.26, 9.39, and 22.48, respectively; Tukey’s test, *q*_4,3_ = 6.82, *p* < 0.05, *n* = 3; Figure 1C). Moreover, the germination rates of YGSL and NIS-001 differed significantly at 25 °C (*q* = 8.23, Tukey’s test, *q*_4,3_ = 6.82, *p* < 0.05, *n* = 3; Figure 1C). Meanwhile, no temperature dependence was observed for the germination rate of any strain.

### 3.4. Germination Rates in KCl + H_2_O_2_ Solution

The germination rates of microsporidian strains were also measured at different temperatures using KCl + H_2_O_2_ solution (Table 2). The mean germination rates of FOA were 70.3% and 77.7% at 4 °C and 25 °C, respectively, compared to 59.3% and 50.0%, respectively, for YGSL, 34.1% and 37.6%, respectively, for OSL, and 84.0% and 77.6%, respectively, for NIS-001. However, the results of NIS-001 obtained for the germination rates varied from 70.6% to 93.3% and from 55.3% to 91.8% at 4 °C and 25 °C, respectively. The germination rates of FOA and YGSL were stable at both temperatures compared to those of NIS-001 in all three trials.

Statistical analysis only revealed significant differences in the mean germination rates among the microsporidian strains (*F*_3,16_ = 3.24, *p* < 0.001; two-way ANOVA, *p* < 0.05, *n* = 3). Multiple comparisons revealed significant differences in the germination rates between FOA and OSL at both test temperatures (*q* = 9.90 and 6.82, respectively) and between YGSL and OSL at 4 °C (*q* = 6.89; Tukey’s test, *q*_4,3_ = 6.82, *p* < 0.05, *n* = 3; Figure 1D).

## 4. Discussion

Chemical and physical stimuli are involved in the germination of microsporidian polar tubes, and the substances that stimulate their germination vary among species and strains [8,19,20,21,22,23,24]. For example, *Nosema apis* and *Paranosema locustae* germinate in the presence of sodium ions [24,25]; however, these microsporidian spores do not germinate in physiological salt solution [26]. Conversely, Nakamura et al. [6] found that OSL germinates after exposure to physiological salt solution. Thus, microsporidian spores may have unique properties, depending on the species and strain.

In this study, we investigated the germination of FOA and YGSL at two different temperatures and in four different treatment solutions. FOA displayed no germination in DW, as also observed for NIS-001. FOA displayed significant differences from OSL in germination, illustrating differences between two strains in the same genus (*Trachipleistophora*). KOH solution was less likely to promote the germination of these strains compared to the findings for NIS-001. In KCl + H_2_O_2_ solution, the germination rate of FOA was approximately twofold higher than that of OSL. The germination of FOA in physiological salt solution was particularly affected by the temperature, as indicated by a 10-fold difference. These results indicated that FOA has different germination characteristics from those of known microsporidian strains of the same genus, including OSL. Meanwhile, FOA belongs to the genus *Trachipleistophora,* according to a classification based on SSU rRNA gene sequences [13], and this strain has a high genetic homology to OSL. Therefore, the attribution of FOA is unclear, and future comparisons with other microsporidia of the same genus are needed.

YGSL had a higher germination rate than OSL and NIS-001 in DW at 25 °C. Furthermore, YGSL exhibited significant differences from NIS-001, with germination rates of 40%–50% recorded in physiological salt solution at 25 °C. The germination of YGSL in DW and physiological salt solution varied by 5–10-fold with an increasing temperature. In contrast, the remaining three strains were less affected by temperature. This suggests that the test temperature has a strong effect on the germination of YGSL. Moreover, these findings indicate that the germination characteristics of YGSL differ from those of *Trachipleistophora* and *Nosema* genera. In previous studies, *Vavraia culicis* and *Vavraia oncoperae* germinated in KCl solution [27,28]. In this study, 50% of YGSL spores germinated in KOH and KCl + H_2_O_2_ solutions containing potassium and chloride ions. Additionally, this strain germinated in DW and the physiological salt solution, with a 20–50% germination rate observed at 25 °C. These characteristics were not observed in the remaining three strains. Therefore, a detailed search for the optimal germination conditions is needed to identify the factors affecting the germination of strains in the genus *Vavraia*.

Protecting silkworms from pébrine and microsporidiosis is necessary for the harvest of excellent cocoons [29,30]. With the exception of our previous report [6], there are no reports on the germination of microsporidia infecting silkworms in physiological salt solution. The present study suggests that the microsporidia of FOA and YGSL, similar to those of OSL, germinate in physiological salt solution. The germination of microsporidia in physiological salt solution may occur in nature. In the future, many microsporidia that cannot be controlled using conventional methods might be detected.

Germination characteristics are important for the use of microsporidia as integrated pest management (IPM) agents and biological pesticides; i.e., the microsporidian spores must be stable and easy to store, even when formulated. In this study, storage in DW at 4 °C was appropriate for FOA, and storage in DW or physiological salt solution at 4 °C was appropriate for YGSL. In both cases, increasing the temperature improved the germination rate. This characteristic can be useful for converting microsporidia into IPM agents. Conversely, the infectivity of these two strains in beneficial insects, such as silkworms, must also be considered. Therefore, actual inoculation experiments on silkworms and agricultural pests must be conducted to verify the toxicity and effectiveness of microsporidia.

## Figures and Tables

**Figure 1 insects-14-00185-f001:**
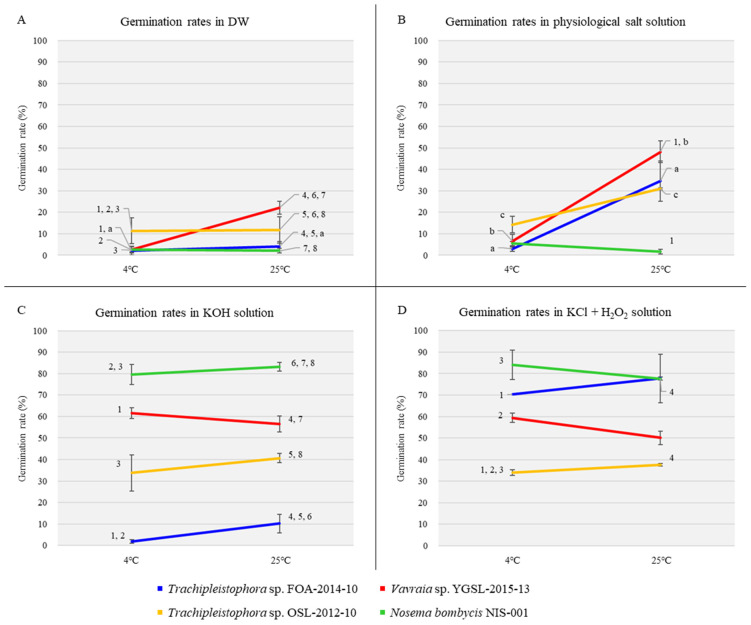
Comparison of the microsporidian spore germination rates in each solution. (**A**) shows the germination rates in DW. The significance of differences was assessed using Bonferroni test (*p* < 0.05). (**B**) shows the germination rates in physiological salt solution. The significance of differences was assessed using Tukey’s test (*p* < 0.05). (**C**) shows the germination rates in KOH solution. The significance of differences was assessed using Tukey’s test (*p* < 0.05). (**D**) shows the germination rates in KCl + H_2_O_2_ solution. The significance of differences was assessed using Tukey’s test (*p* < 0.05). The same numbers (1–8) or letters (a–c) indicate significant differences between strains or test temperatures, respectively (*p* < 0.05). Data on the germination rates of *Trachipleistophora* sp. OSL-2012-10 and *Nosema bombycis* NIS-001 were quoted from Nakamura et al. [6]. The bars represent standard errors.

**Table 1 insects-14-00185-t001:** List of microsporidian strains.

Strain	*Trachipleistophora* sp. FOA-2014-10	*Vavraia* sp. YGSL-2015-13
Abbreviation	FOA	YGSL
Spore size (µm)	4.42 ± 0.28 × 2.35 ± 0.14	5.36 ± 0.40 × 2.91 ± 0.20
Isolation host	*Orthetrum albistylum speciosum*	*Spodoptera litura*
Capture area	Kanagawa, Japan	Yamaguchi, Japan
Capture year	2014	2015
Reference	Nakamura et al. [13]	Imura et al. [14]
Strain	*Trachipleistophora* sp. OSL-2012-10	*Nosema bombycis* NIS-001
Abbreviation	OSL	NIS-001
Spore size (µm)	4.54 ± 0.38 × 2.66 ± 0.22	3.6 × 2.2
Isolation host	*Spodoptera litura*	*Bombyx mori*
Capture area	Tokyo (Ogasawara), Japan	—
Capture year	2012	—
Reference	Shigano et al. [12]; Nakamura et al. [6]	Fujiwara [17]

**Table 2 insects-14-00185-t002:** List of the germination rates of microsporidia strains. The numbers in parentheses regarding the germination rates indicate the standard error. NaCl denotes physiological salt solution. Data on the germination rates of *Trachipleistophora* sp. OSL-2012-10 and *Nosema bombycis* NIS-001 were taken from Nakamura et al. [6].

Strain	Condition	Germination Rates (%)							
		4 °C				25 °C			
		1st	2nd	3rd	Mean	1st	2nd	3rd	Mean
*Trachipleistophora* sp.	DW	0.0 (±0.9)	0.0 (±1.3)	6.1 (±0.4)	2.0 (±2.0)	8.3 (±3.9)	1.3 (±0.9)	2.8 (±1.3)	4.1 (±2.1)
FOA-2014-10	NaCl	4.7 (±1.9)	1.1 (±1.1)	3.0 (±0.6)	2.9 (±1.0)	52.9 (±2.3)	28.6 (±1.3)	22.2 (±1.6)	34.6 (±9.4)
	KOH	1.8 (±0.8)	0.6 (±0.6)	3.2 (±1.0)	1.9 (±0.8)	18.9 (±3.1)	6.4 (±0.7)	5.4 (±0.5)	10.2 (±4.3)
	KCl + H_2_O_2_	70.3 (±1.1)	70.3 (±0.4)	70.3 (±0.2)	70.3 (±0.0)	76.7 (±2.3)	79.2 (±2.1)	77.2 (±2.9)	77.7 (±0.8)
*Vavraia* sp.	DW	0.0 (±7.9)	5.5 (±6.3)	2.2 (±2.3)	2.6 (±1.6)	20.5 (±5.7)	28.1 (±2.1)	18.0 (±4.2)	22.2 (±3.0)
YGSL-2015-13	NaCl	9.5 (±2.7)	10.1 (±4.7)	0.0 (±4.1)	6.5 (±3.3)	55.6 (±5.1)	50.2 (±6.7)	38.8 (±5.2)	48.2 (±5.0)
	KOH	59.3 (±1.6)	66.7 (±2.4)	58.8 (±3.4)	61.6 (±2.6)	61.3 (±0.9)	59.2 (±4.3)	49.0 (±0.8)	56.5 (±3.8)
	KCl + H_2_O_2_	57.6 (±0.8)	63.6 (±5.7)	56.7 (±2.6)	59.3 (±2.2)	48.6 (±2.5)	55.9 (±2.7)	45.6 (±2.2)	50.0 (±3.1)
*Trachipleistophora* sp.	DW	20.2 (±0.8)	0.0 (±5.2)	14.0 (±2.3)	11.4 (±6.0)	20.3 (±3.1)	15.1 (±1.5)	0.0 (±2.7)	11.8 (±6.1)
OSL-2012-10	NaCl	20.5 (±0.7)	15.3 (±1.7)	7.0 (±2.4)	14.3 (±3.9)	31.9 (±4.1)	30.6 (±5.8)	30.4 (±3.6)	31.0 (±0.5)
	KOH	46.4 (±2.5)	17.9 (±3.6)	37.0 (±1.2)	33.8 (±8.4)	36.5 (±2.1)	41.6 (±1.6)	44.0 (±4.6)	40.7 (±2.2)
	KCl + H_2_O_2_	36.4 (±3.3)	31.8 (±8.6)	34.1 (±2.9)	34.1 (±1.3)	36.6 (±3.0)	38.3 (±4.7)	38.0 (±2.2)	37.6 (±0.5)
*N. bombycis*	DW	5.9 (±1.1)	0 (±1.4)	1.7 (±1.7)	2.5 (±1.8)	2.2 (±2.1)	3.9 (±2.1)	0 (±0.3)	2.1 (±1.1)
NIS-001	NaCl	6.9 (±1.2)	3.5 (±0.5)	5.9 (±0.6)	5.4 (±1.0)	1.2 (±1.2)	3.6 (±1.4)	0 (±0.5)	1.6 (±1.1)
	KOH	88.7 (±1.5)	72.8 (±2.6)	77.2 (±2.0)	79.6 (±4.7)	79.5 (±3.0)	86.3 (±0.9)	83.8 (±3.5)	83.2 (±2.0)
	KCl + H_2_O_2_	70.6 (±2.3)	93.3 (±1.2)	88.0 (±1.8)	84.0 (±6.9)	55.3 (±4.2)	91.8 (±1.2)	85.8 (±0.5)	77.6 (±11.3)

## Data Availability

Data sharing not applicable.

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
