# Peer review of "Comparison of the Germination Conditions of Two Large-Spore Microsporidia Using Potassium and Sodium Ion Solutions"

_insects, 2023, doi:10.3390/insects14020185_

Round 1
Reviewer 1 Report
The authors investigated the germination conditions of newly two isolated microsporidia, and showed that germination conditions the the microsporidia species vary depending on the solution and temperature at which they are stored. The results are interesting and provide necessary knowledge for the study of microsporidia, especially as they show a high germination rate at 25°C even in DW. For this reason, this paper is worthy of publication, but needs significant revision, including in English. The MS should be edited by a native English speaker, and the authors themselves should thoroughly review the paper. Many easy errors were found, for example, the species name of Nosema bonbycis in Simple summry is not italicized, and the plural and singular forms were found in a sentence.
 In addition, the description of Material & Method is insufficient: p. 96 "The conditions were as described below," but no description of the experiment is given. There is no description of the temperatures (4°C and 25°C) and the duration of the test, and the repetition of the experiment.
 There is also no description of the statistical analysis or the software used. Furthermore, the authors used Tukey/Bonferroni, there is no description of test for normality. In addition, please provide statistics for the results.
Other comments
L15 "Nosema bombycis" -> "Nosema bombycis"
L21 "hatch" -> "hatched"
L54-55 In the text, the subject is “insects”, but the verb is a description of microsporida
L66-68 Can the authors say that microsporidia can enter a silkworm rearing room simply by the fact that microsporidia of the same size were found in the field? What is the evidence for this?
L90 What does "were taken from Nakamura et al." mean? Is this meaning that the two species of microsporidia reported by Nakamura were used?
L138-156 In this section, the expressions “rate” and “was” and “rates” and “were” were present. Shouldn't they all be plural?
L204-207 Start a new line form "In this study,"
L221 "Therefore" -> "Although"
L233 "V. cilicis" should be “Vavraia culicis” because it is not mentioned before.
L246-247 Grammar is incorrect.
L250 "that isa," -> "that is,"
Author Response
Response to Reviewer 1 Comments
Point 1: The authors investigated the germination conditions of newly two isolated microsporidia, and showed that germination conditions the the microsporidia species vary depending on the solution and temperature at which they are stored. The results are interesting and provide necessary knowledge for the study of microsporidia, especially as they show a high germination rate at 25°C even in DW. For this reason, this paper is worthy of publication, but needs significant revision, including in English. The MS should be edited by a native English speaker, and the authors themselves should thoroughly review the paper. Many easy errors were found, for example, the species name of Nosema bonbycis in Simple summry is not italicized, and the plural and singular forms were found in a sentence.
In addition, the description of Material & Method is insufficient: p. 96 "The conditions were as described below," but no description of the experiment is given. There is no description of the temperatures (4°C and 25°C) and the duration of the test, and the repetition of the experiment.
There is also no description of the statistical analysis or the software used. Furthermore, the authors used Tukey/Bonferroni, there is no description of test for normality. In addition, please provide statistics for the results.
Response: We thank you for providing these insightful comments. As described in Section 2.2., we followed the experimental protocol reported by Nakamura et al. [6]. However, it was not sufficiently described. Hence, we have added the details of the experimental procedure.
For statistical analysis, Excel was used for calculations. Because the sample size in this study was small, statistical analysis was conducted assuming that normality and homoscedasticity were followed. We have clarified the statistical tests in the Results section.
Point 2: L15 "Nosema bombycis" -> "Nosema bombycis"
Response: We apologize for the error, which has been corrected.
Point 3: L21 "hatch" -> "hatched"
Response: We have corrected the text to “germinated”.
Point 4: L54-55 In the text, the subject is “insects”, but the verb is a description of microsporida
Response: We agree with your comment and have revised the sentence accordingly.
Point 5: L66-68 Can the authors say that microsporidia can enter a silkworm rearing room simply by the fact that microsporidia of the same size were found in the field? What is the evidence for this?
Response: We have revised the text to clarify our intent.
Point 6: L90 What does "were taken from Nakamura et al." mean? Is this meaning that the two species of microsporidia reported by Nakamura were used?
Response: We have rephrased the sentence to clarify that we quoted the germination data from Nakamura et al.
Point 7: L138-156 In this section, the expressions “rate” and “was” and “rates” and “were” were present. Shouldn't they all be plural?
Response: We apologize for these errors, which have been corrected throughout the manuscript.
Point 8: L204-207 Start a new line form "In this study,"
Response: We have made the change as you indicated.
Point 9: L221 "Therefore" -> "Although"
Response: We have made the change as you indicated.
Point 10: L233 "V. cilicis" should be “Vavraia culicis” because it is not mentioned before.
Response: We apologize for this error, which has been corrected.
Point 11: L246-247 Grammar is incorrect.
Response 11: We have rectified grammatical errors in the text.
Point 12: L250 "that isa," -> "that is,"
Response: We apologize for this error, which has been corrected.

Reviewer 2 Report
Imura et al. entitled “Comparison of the Germination Conditions of Two Large spore Microsporidia using Potassium and Sodium Ion Solutions” that reported the germination conditions of two types microsporidia. However, there are still some issues that need to be clear before it is published.
Major revision
1. Figure 1 and Table 1 did not show any significantly difference. Actually, it looks demonstrate the similar results and instead of morphology of spores.
2. Author can display the result of morphology of spores in different conditions.
Author Response
Response to Reviewer 2 Comments
Point 1: 1. Figure 1 and Table 1 did not show any significantly difference. Actually, it looks demonstrate the similar results and instead of morphology of spores.
Response: Are you referring to Figure 1 and Table 2? Figure 1 presents significant differences using numbers and letters. Standard errors have also been added in response to the comments provided by Reviewer 1. In addition, Table 2 provides the data of Figure 1 in a numerical format, and it can be deleted if considered unnecessary.
Point 2: 2. Author can display the result of morphology of spores in different conditions.
Response: We havo not captured the microsporodia spores observed in different conditions, as there is no difference in conditions. Therefore, we cannot provide the results of morphology of spores in different conditions.

Reviewer 3 Report
I had trouble understanding the Materials and Methods. There are too many repetitions of the species- and their number names. I would suggest introducing all species names once, then showing all species names in a table with abbreviations, then use abbreviations in the text.
In Table 1 there are two species mentioned, in the results there are 4 species compared, why?
Results: Please give the results in line graphs with Standard errors of the mean. Y-Axis = Species, X-Axis is temperature
Or show the same species germination at different temperatures (x-axis) and saline solutions on y-axis.
You had N= 3 per trial, so show standard errors of the mean.
Also: Were the data normally distributed, so you were able to use an ANOVA?
Were the samples independent from each other?
Please provide more information about the Statistical analysis.
Please do a check of the document to get rid of typos, the English is otherwise fine, I think.
Good luck with your work
Author Response
Response to Reviewer 3 Comments
Point 1: I had trouble understanding the Materials and Methods. There are too many repetitions of the species- and their number names. I would suggest introducing all species names once, then showing all species names in a table with abbreviations, then use abbreviations in the text.
Response: We have abbreviated the species names as follows:
Trachipleistophora sp. FOA-10 (FOA)
Vavraia sp. YGSL-2015-13 (YGSL)
Trachipleistophora sp. OSL-2012-10 (OSL)
N. bombycis NIS-001 (NIS-001)
Point 2: In Table 1 there are two species mentioned, in the results there are 4 species compared, why?
Response: We abbreviated OSL-2012-10 and NIS-001 in Table 1 because they had been previously reported. In response to your suggestion, we have included all strains in Table 1.
Point 3: Results: Please give the results in line graphs with Standard errors of the mean. Y-Axis = Species, X-Axis is temperature. Or show the same species germination at different temperatures (x-axis) and saline solutions on y-axis. You had N= 3 per trial, so show standard errors of the mean.
Response: We have revised the figure as suggested and added the standard deviations.
Point 4: Also: Were the data normally distributed, so you were able to use an ANOVA? Were the samples independent from each other?
Response 4: We performed our statistical analysis assuming normality because of the small sample size. In addition, the samples were independent.
Point 5: Please provide more information about the Statistical analysis.
Response: In two-way ANOVA, we performed statistical analyses using “Analysis tool” included with the add-in in Excel. In multiple comparisons, we performed statistical analysis by entering the formulae into Excel.

Round 2
Reviewer 1 Report
No more comment on the MS.
Author Response
Response to Reviewer 1 Comments
Thanks for reviewing.
English proofreading was orderd by enago.
Reviewer 3 Report
The paper has improved strongly by the revisions.
I think it is good for publication.
One small thing: Add N's for every description of statistical analysis, just add them into the parenthesis whenever you write the p-values and F-Values in in the text.
Author Response
Response to Reviewer 3 Comments
English proofreading was ordered by enago.
Point1:One small thing: Add N's for every description of statistical analysis, just add them into the parenthesis whenever you write the p-values and F-Values in in the text.
Response
Added N's for every description of statistical analysis.